# Discovery of [1,2,4]Triazole Derivatives as New Metallo-β-Lactamase Inhibitors

**DOI:** 10.3390/molecules25010056

**Published:** 2019-12-23

**Authors:** Chen Yuan, Jie Yan, Chen Song, Fan Yang, Chao Li, Cheng Wang, Huiling Su, Wei Chen, Lijiao Wang, Zhouyu Wang, Shan Qian, Lingling Yang

**Affiliations:** 1College of Food and Bioengineering, Xihua University, Chengdu 610039, China; 18380460102@163.com (C.Y.); 18328072545@163.com (J.Y.); sc1475778467@163.com (C.S.); 18228209592@163.com (F.Y.); 18408240706@163.com (C.L.); 15182011493@163.com (C.W.); 15982178237@163.com (H.S.); CW13980419963@163.com (W.C.); wanglijiao@mail.xhu.edu.cn (L.W.); 2College of Science, Xihua University, Chengdu 610039, China; zhouyuwang77@gmail.com

**Keywords:** β-lactam resistance, metallo-β-lactamase, serine β-lactamase, VIM-2, triazole

## Abstract

The emergence and spread of metallo-β-lactamase (MBL)-mediated resistance to β-lactam antibacterials has already threatened the global public health. A clinically useful MBL inhibitor that can reverse β-lactam resistance has not been established yet. We here report a series of [1,2,4]triazole derivatives and analogs, which displayed inhibition to the clinically relevant subclass B1 (Verona integron-encoded MBL-2) VIM-2. 3-(4-Bromophenyl)-6,7-dihydro-5*H*-[1,2,4]triazolo [3,4-*b*][1,3]thiazine (**5l**) manifested the most potent inhibition with an IC_50_ (half-maximal inhibitory concentration) value of 38.36 μM. Investigations of **5l** against other B1 MBLs and the serine β-lactamases (SBLs) revealed the selectivity to VIM-2. Molecular docking analyses suggested that **5l** bound to the VIM-2 active site via the triazole involving zinc coordination and made hydrophobic interactions with the residues Phe61 and Tyr67 on the flexible L1 loop. This work provided new triazole-based MBL inhibitors and may aid efforts to develop new types of inhibitors combating MBL-mediated resistance.

## 1. Introduction

The β-lactams are the most widely used antibacterial agents in clinical practice for many years [1]. However, β-lactam resistance has become a major threat to global public health due to the emergence and rapid spread of β-lactam-resistant bacterial pathogens [2,3,4,5]. One of the important mechanisms of resistance to β-lactam antibiotics is the production of β-lactamases that can hydrolyze the core β-lactam ring by a nucleophilic reaction to inactive the drugs [6,7]. According to the catalytic mechanisms, β-lactamases are grouped into two catalogs: serine β-lactamases (SBLs, using the active site serine residue as a nucleophile) and metallo-β-lactamases (MBLs, using the Zn^2+^ activated hydroxide as a nucleophile) [6,7,8]. Up to now, to circumvent bacterial resistance mediated by β-lactamases, five clinically useful SBL inhibitors have been developed, including clavulanic acid, sulbactam, tazobactam, avibactam, and vaborbactam [9,10,11,12]. In contrast, there remain no FDA-approved small-molecule inhibitors for MBLs, which can hydrolyze almost all β-lactam antibiotics, including the last-generation cephalosporins and carbapenems [13,14]. MBL enzymes are divided into three subclasses: B1, B2, and B3 [7]. Among them, B1 MBLs are the most clinically relevant, typically including New Delhi MBLs (NDMs), Verona integron-encoded MBLs (VIMs), and imipenemases (IMPs) [15].

To date, there have been a number of small-molecule MBL inhibitors reported, as exemplified in Figure 1. The thiol-based compounds (e.g., L-Captopril, **II** and **III**) were widely investigated, which had potent inhibitory activities against MBL enzymes, including subclass B1 VIM-2 and NDM-1 [16,17,18,19,20,21,22,23]. The compounds **IV** and **V**, containing a triazole moiety, displayed a relatively weak inhibitory potency to B1 VIM-2 and B3 L1 MBL [24,25,26]. The 3-oxoisoindoline-4-carboxylic acids (**VI**) and 4-hydroxyisoquinoline-3-carbonyls (**VII**) have a selective inhibition to B1 VIM-2 and VIM-5 enzyme, respectively, which were observed by crystallographic analyses to bind with the active site but did not chelate with the zinc ions [27,28]. The phthalic acid derivative (**VIII**) inhibits the IMP-1 enzyme with an IC_50_ (half-maximal inhibitory concentration) value of 2.7 μM [29]. Cyclic boronate compound **IX** (taniborbactam) exhibits potent inhibitory activities against multiple MBL enzymes, including B1 VIM-1 (IC_50_ = 0.0079 μM), B1 VIM-2 (IC_50_ = 0.0005 μM), and B1 NDM-1 (IC_50_ = 0.01 μM), via a mechanism involving mimicking of their common natural tetrahedral intermediates [21,30,31]. Furthermore, some natural products (e.g., X, AMA, and rosmarinic acid) and their derivatives were reported to have moderate inhibitory activities against MBLs [8,32,33]; AMA is a rapid and potent inhibitor of NDM-1 and VIM-2, which works through a metal-stripping manner and can markedly restore the activity of meropenem against bacteria producing NDM-1 and VIM-2 in vitro and in vivo [8,34]. Nevertheless, currently, there is a need to develop new MBL inhibitors to provide more hit/lead compounds for structural optimization and drug development.

By random screening of our in-house compound library, we identified some hit compounds with moderate inhibitory activity to B1 VIM-2. Among them, 3-phenyl-6,7-dihydro-5*H*-[1,2,4]triazolo [3,4-b][1,3]thiazine (**5a**, Figure 2) displayed about 50% inhibition to VIM-2 at 100 μM (IC_50_ ~ 179 μM, the inhibition curve please see Appendix A), and its chemical scaffold has not been reported as an MBL inhibitor. A series of **5a** derivatives and analogs was hence synthesized (Figure 2) and tested for enzyme inhibition capabilities. The most potent compounds were also investigated for their selectivity and possible binding modes.

## 2. Results and Discussion

### 2.1. Chemistry

The synthetic routes and total yields of all target compounds are outlined in Scheme 1, Scheme 2, Scheme 3. Compounds **5a**–**5t** and **6** were synthesized using the reaction sequence shown in Scheme 1. Different carboxylic acids (**1a**–**1t**) were used as the staring materials, and the corresponding esters (**2a**–**2t**) were readily synthesized by reactions of the acids with MeOH in the presence of sulfurous dichloride. The resulting esters **2a**–**2t** were reacted with hydrazine hydrate to produce the corresponding hydrazides (**3a**–**3t**), followed by reacting with ammonium thiocyanate to afford the 2,4-dihydro-3*H*-1,2,4-triazole-3-thiones (**4a**–**4t**) in high yields. Subsequently, the **4a**–**4t** were reacted with 1,3-dibromopropane or 1,2-dibromoethane in the presence of NaOH and NaHCO_3_ at 80 °C for 6 h to give the desired target compounds **5a**–**5t [35]**. Compound **5d** was subjected to the demethoxyl reaction by boron tribromide to give the final compound **6** in 58% yield.

Scheme 2 shows the synthetic route for compounds **10a** and **10b**. Furan-2-carbonyl chloride (**7a**) or thiophene-2-carbonyl chloride (**7b**) was reacted with hydrazine hydrate to afford the corresponding hydrazides **8a** or **8b**, respectively. Next, **8a** or **8b** was under cyclization reaction by a method similar to the synthesis of **4a**–**4t** in the presence of 10% NaOH, to produce 2,4-dihydro-3*H*-1,2,4-triazole-3-thiones **9a** or **9b**. Finally, the target compounds **10a** and **10b** were obtained in excellent yields by the reactions of **9a** or **9b** with 1,3-dibromopropane.

The synthesis of compounds **14** and **15**, which contain a carboxyl group substituted at the C2 position of benzene ring of 3-phenyl-6,7-dihydro-5*H*-[1,2,4]triazolo[3,4-*b*][1,3]thiazine or 3-phenylthiazolo[2,3-c][1,2,4]triazol-5(6*H*)-one, are depicted in Scheme 3. Reaction of phthalic anhydride (**11**) with hydrazinecarbothioamide gave intermediate **12** at 80 °C for 4 h. Compound **13** was then prepared by ring closure reaction of intermediate **12** in the presence of 10% NaOH with 72% yield [36]. The target compound **14** was finally synthesized from **13** using a method similar to that for **10a** and **10b**. The target compound **15** was prepared by treatment of the key intermediate **13** with 2-bromoacetic acid in the presence of NaAc at room temperature for 0.5 h and then 80 °C for 6 h in EtOH. All the synthesized target compounds (**5a**–**5t**, **6**, **10a**, **10b**, **14,** and **15**) were subjected to NMR (for the spectrogram, please see Appendix A), ESI-MS, and HPLC analyses for their structure and purity confirmation.

### 2.2. SAR of [1,2,4]Triazole Derivatives Using Enzyme Inhibitory Assays

The inhibitory activities of these target compounds were first tested against B1 VIM-2 at concentrations of 100 and 10 μM. Compared with the hit compound **5a**, compounds **5b**–**5i**, **6**, and **14**, containing an *ortho* or *meta* position substituents on the phenyl ring, showed comparable or slightly lower potency to VIM-2 at 100 μM, except for **5i** (52% ± 4%) and **6** (57% ± 5%) (Table 1). Different halogen (F, Cl, Br, and I) or trifluoromethyl moiety substitutes the *para* position of phenyl group yielding the compounds **5j**–**5n**, respectively. With the exception of **5j** (with F at *para*-phenyl), compounds **5k**–**5n** displayed more potent inhibition against VIM-2 at 100 or 10 μM than unsubstituted **5a**. However, a larger substituent (*t*-butyl moiety) at the *para* position of phenyl (**5o**) showed significantly decreased activity to VIM-2. Next, we examined the possible influence of disubstitution (**5p**) on phenyl group and 4-acetamido-aniline substitution at the *para* position of phenyl group (**5t**) (Table 1). Both compounds **5p** and **5t** exhibit considerable potency against VIM-2, with the inhibition rate of 71% ± 6%/42% ± 5% and 75% ± 4%/40% ±3% at 100 μM/10 μM, respectively. Compounds **5q**, **5r**, **10a**, and **10b**, with 2-pyridyl (**5q**), benzyl (**5r**), 2-furanyl (**10a**), and 2-thienyl (**10b**) replacing phenyl (**5a**), also showed decreased activities against VIM-2 (Table 1). Compared with 6,7-dihydro-5*H*-[1,2,4]triazolo[3,4-b][1,3]thiazine scaffold, 5,6-dihydrothiazolo[2,3-c][1,2,4]triazole (5s vs. 5l) and thiazolo[2,3-c][1,2,4]triazol-5(6*H*)-one (**15** vs. **14**) seemed to be less favorable for the inhibitory activities against VIM-2, for example **5l** (86% ± 5%/53% ± 4%) vs. **5s** (80% ± 3%/50% ± 2%), and **14** (26% ± 2%/21% ± 2%) vs. **15** (21% ± 2%/18% ± 3%) (Table 1).

Then, we tested all the target compounds against other B1 MBL enzymes, including NDM-1, IMP-1, VIM-1, and VIM-5 (Table 1); all the assay conditions (including enzyme/substrate concentrations) are the same as that previously used [12,23]. We observed that all of them exhibited relatively weak ability to inhibit these enzymes compared with VIM-2. Among these compounds, 3-(4-(tert-butyl)phenyl)-6,7-dihydro-5*H*-[1,2,4]triazolo[3,4-b][1,3]thiazine (5o) and 3-(thiophen-2-yl)-6,7-dihydro-5*H*-[1,2,4]triazolo[3,4-b][1,3]thiazine (10b) displayed more than 50% inhibition toward IMP-1 at 100 μM (63% ± 1% and 57% ± 6%, respectively). Moreover, compound **5n**, with a trifluoromethyl moiety at the *para* position of the phenyl group, showed promising potency with 61% ± 3% VIM-1 inhibition at 100 μM. Nevertheless, compounds **5o**, **10b,** or **5n** only have limited activity against IMP-1 or VIM-1 and need further optimization for these MBL types.

The preliminary SAR studies led to the discovery of a number of compounds that exhibited more potent inhibition against MBLs than the hit compound **5a**. For these compounds (>50% inhibition rate against the corresponding targets), we then further performed dose–response studies (i.e., half-maximal inhibitory concentration, IC_50_) against the corresponding targets, and the results are presented in Figure 3 and Figure 4. As shown in Figure 3, compounds **5k**, **5l**, **5n**, **5p**, and **5s** both inhibit VIM-2 in a dose-dependent manner with the IC_50_ values less than 100 μM; and the IC_50_ values for **5k**, **5l**, **5n**, **5p**, and **5s** are 47.24, 38.36, 53.20, 53.85, and 67.16 μM, respectively. Figure 4 shows the IC_50_ curves of **5o** against IMP-1, **5n** against VIM-1, and **10b** against IMP-1. Obviously, these three compounds did not have potent inhibition to these tested MBLs (IC_50_ > 100 μM). The most potent compound (**5l**) was hence chose to perform selectivity investigation and binding mode prediction.

Considering that MBLs and SBLs are two catalogs of β-lactamases, we further tested the compound **5l** against some representative SBL enzymes, including KPC-2 (Klebsiella pneumoniae carbapenemase 2), TEM-1, AmpC, and OXA-48 (Oxacillinase 48), with the aim of investigating its selectivity; particularly, this is used as a counter screening to indicate the specific inhibition to MBLs. No or low inhibitory activities to KPC-2, TEM-1, and OXA-48 were observed for **5l** even at 100 μM (Table 2). Relatively, compound **5l** displayed only weak inhibition (about 50% inhibition at 100 μM) to AmpC. Together, these results suggest that **5l** is a selective VIM-2 MBL inhibitor.

The molecular docking analysis was then used to investigate the possible binding mode of **5l** with VIM-2. A total of 10 possible binding modes was generated by using GOLD and AutoDock Vina program. No significant difference was observed for the binding modes predicted by these two programs. The top docking pose (with Goldscore of 53.18, and Vinascore of −7.5 kcal/mol) was considered as the most possible binding mode, as shown in Figure 5. We observed that **5l** likely bound with the active site of VIM-2 in a metal coordination manner (Figure 5) via the triazole moiety that has been reported as a metal-binding pharmacophore to coordinate with MBL enzymes (e.g., 5ACW) and other zinc metalloenzymes [12]. The triazole of **5l** is likely positioned to form a coordination bond with the active site Zn1; the distance between the nitrogen atom of triazole and Zn1 is about 2.5 Å (Figure 5a). Compound **5l** is also likely placed to make hydrophobic interactions with the residues Tyr67 and Phe61 (using the standard BBL (class B β-lactamases) numbering scheme for class B β-lactamases) on the flexible L1 loop; notably, the phenyl group appears to form π–π stacking interactions with Tyr67 [37]. Moreover, the phenyl of **5l** likely has interactions with the residue Arg228, which is important for the recognition of β-lactam carboxylate.

## 3. Materials and Methods

### 3.1. Synthesis

All solvents were analytical reagents (ARs), commercially available, and used without further purification. The reaction systems were monitored by thin-layer chromatography with silica gel precoated glass and fluorescent indicator, and the removal of solvent was carried out with a rotary evaporator and vacuum pump. As previously reported, proton (^1^H) and carbon (^13^C) NMR spectra were recorded on a Bruker AV-400 instrument and are reported in ppm relative to tetramethylsilane (TMS) and referenced to the solvent in which the spectra were collected. Low-resolution and high-resolution mass spectral (MS) data were determined on an Agilent 1100 Series LC-MS with UV detection at 254 nm and a low-resonance electrospray mode (ESI). All target compounds were purified to >95% purity, as determined by high-performance liquid chromatography (HPLC). The HPLC analysis was performed on a Waters 2695 HPLC system equipped with a Kromasil C18 column (4.6 mm × 250 mm, 5 μm).

#### General Procedure 1: SOCl_2_-Mediated Ester Formation

To a solution of 5.0 mmol of different substituted benzoic acids (**1a**~**1p**, **1s**, and **1t**), nicotinic acid (**1q**) or 2-phenylacetic acid (**1r**) in MeOH (15 mL) was added sulfurous dichloride (2 mL), then the mixture was allowed to reach 40 °C and stirred for 1 h. The resulting solution was concentrated and partitioned between sodium bicarbonate solution (PH 7~8) and ethyl acetate (3×). The organic layer was dried over anhydrous magnesium sulfate, filtered, and concentrated to yield the corresponding esters (**2a**~**2t**) in 89%–96% yields, which were taken up for the next step without any purification.

#### General Procedure 2: The Formation of Hydrazide

To a solution of esters (**2a**~**2t**, 1.0 equiv.), furan-2-carbonyl chloride (**7a**, 1.0 equiv.) or thiophene-2-carbonyl chloride (**7b**, 1.0 equiv.) in MeOH (2 mL/1 mmol) was added hydrazine hydrate (1 mL/1 mmol), then the mixture was allowed to reach 65 °C and stirred for 4 h. After completion (monitored by TLC), the organic solvent was removed and extracted three times with ethyl acetate, the combined organic extracts were dried (Na_2_SO_4_) and concentrated under reduced pressure to give the corresponding hydrazides (**3a**~**3t**, **8a**, or **8b**) in high yields, which were taken up for the next step without any purification.

#### General Procedure 3: Ammonium Thiocyanate-Involved Ring Closing Reaction

To a solution of hydrazides (**3a**~**3t**, **8a**, or **8b**, 1.0 equiv.) 10% NaOH aqueous solution (1.5 mL/1 mmol) was added ammonium thiocyanate (3.0 equiv.), then the mixture was heated to 80 °C and stirred for 6 h. The mixture was then cooled to room temperature and filtered. The filtrate was neutralized to pH 3–4 by concentrated hydrochloric acid, and the resulting white solid was collected by filtration. The combined filter cake was dried to give the 2,4-dihydro-3*H*-1,2,4-triazole-3-thiones in 72%–83% yields.

#### General Procedure 4: Dibromoalkane-Involved Ring Closing Reaction

A mixture of 2,4-dihydro-3*H*-1,2,4-triazole-3-thiones (**4a**~**4t**, **9a**, or **9b**, 1.0 equiv.), NaHCO_3_ (3.0 equiv.), and KOH (1.0 equiv.) in isopropyl alcohol (2 mL/mmol) was stirred at room temperature for 0.5 h. Then, 1,3-dibromopropane or 1,2-dibromoethane (3.0 equiv.) was added, and the mixture was heat to 80 °C and stirred for 6 h. Upon completion of the reaction as determined by TLC, the organic solvent was removed and the residue was purified by column chromatography to give the desired target compounds **5a**~**5t**, **10a**, **10b**, and **14** with yields ranging from 58% to 72%.

The target compounds **5a**~**5t** were obtained by general procedure 1–4 in turn. Their total yields and characterization data are as follows.

*3-Phenyl-6,7-dihydro-5H-[1,2,4]triazolo[3,4-b][1,3]thiazine* (**5a**) 48% yield, 96.8% HPLC purity. ^1^H-NMR (400 MHz, CDCl_3_) δ 8.04 (d, *J* = 7.6 Hz, 2H), 7.58 (t, *J* = 7.2 Hz, 1H), 7.45 (t, *J* = 7.6 Hz, 2H), 4.48 (t, *J* = 6.0 Hz, 2H), 3.56 (t, *J* = 6.4 Hz, 2H), 2.36–2.30 (m, 2H) ppm. ^13^C-NMR (101 MHz, CDCl_3_) δ 165.41, 151.77, 132.12, 129.00, 128.60, 127.44, 61.68, 30.84, 28.49 ppm. ESI-MS *m/z*: 218.1 [M + H]^+^.

*3-(O-Tolyl)-6,7-dihydro-5H-[1,2,4]triazolo[3,4-b][1,3]thiazine* (**5b**) 43% yield, 97.1% HPLC purity. ^1^H-NMR (400 MHz, CDCl_3_) δ 7.83 (dd, *J* = 8.0 Hz, *J* = 1.2 Hz, 1H), 7.34 (td, *J* = 8.0 Hz, *J* = 1.2 Hz, 1H), 7.19–7.15 (m, 2H), 4.37 (t, *J* = 6.0 Hz, 2H), 3.48 (t, *J* = 6.4 Hz, 2H), 2.5 (s, 3H), 2.28–2.21 (m, 2H) ppm. ^13^C-NMR (101 MHz, CDCl_3_) δ 167.37, 140.28, 132.15, 131.79, 130.58, 129.37, 125.77, 62.46, 31.80, 29.60, 21.84 ppm. ESI-MS *m/z*: 232.1 [M + H]^+^.

*3-(2-Bromophenyl)-6,7-dihydro-5H-[1,2,4]triazolo[3,4-b][1,3]thiazine* (**5c**) 45% yield, 96.5% HPLC purity. ^1^H-NMR (400 MHz, CDCl_3_) δ 7.78 (dd, *J* = 7.6 Hz, *J* = 2.0 Hz, 1H), 7.66 (dd, *J* = 7.6 Hz, *J* = 2.0 Hz, 1H), 7.34–7.32 (m, 2H), 4.45 (t, *J* = 6.0 Hz, 2H), 3.58 (t, *J* = 6.0 Hz, 2H), 2.36–2.30 (m, 2H) ppm. ^13^C-NMR (101 MHz, CDCl_3_) δ 166.14, 134.37, 132.67, 132.18, 131.36, 127.23, 121.56, 63.36, 31.64, 29.50 ppm. ESI-MS *m/z*: 296.0, 298.0 [M + H]^+^.

*3-(2-Methoxyphenyl)-6,7-dihydro-5H-[1,2,4]triazolo[3,4-b][1,3]thiazine* (**5d**) 42% yield, 96.2% HPLC purity. ^1^H-NMR (400 MHz, *DMSO-d_6_*) δ 7.67 (d, *J* = 7.6 Hz, 1H), 7.34 (t, *J* = 7.6 Hz, 1H), 7.15(d, *J* = 8.4 Hz, 1H), 7.02 (t, *J* = 7.6 Hz, 1H), 4.31 (t, *J* = 6.0 Hz, 2H), 3.83 (s, 3H), 3.66 (t, *J* = 6.4 Hz, 2H), 2.25–2.19 (m, 2H) ppm. ^13^C-NMR (101 MHz, *DMSO-d_6_*) δ 166.23, 158.60, 133.97, 131.09, 120.88, 120.55, 120.46, 113.02, 62.24, 56.25, 32.04, 31.62 ppm. ESI-MS *m/z*: 248.1 [M + H]^+^.

*3-(3-Fluorophenyl)-6,7-dihydro-5H-[1,2,4]triazolo[3,4-b][1,3]thiazine* (**5e**) 50% yield, 98.0% HPLC purity. ^1^H-NMR (400 MHz, CDCl_3_) δ 7.93 (d, *J* = 8.0 Hz, 1H), 7.71 (dt, *J* = 9.6 Hz, *J* = 2.0 Hz, 1H), 7.46–7.40 (m, 1H), 7.27 (td, *J* = 9.4 Hz, *J* = 2.0 Hz, 1H), 4.48 (t, *J* = 6.0 Hz, 2H), 3.55 (t, *J* = 6.0 Hz, 2H), 2.36–2.30 (m, 2H) ppm. ^13^C-NMR (101 MHz, CDCl_3_) δ 165.28, 163.79, 161.33, 132.20, 130.09, 125.34, 120.14, 116.50, 63.07, 31.73, 29.26 ppm. ESI-MS *m/z*: 236.1 [M + H]^+^.

*3-(3-Chlorophenyl)-6,7-dihydro-5H-[1,2,4]triazolo[3,4-b][1,3]thiazine* (**5f**) 51% yield, 97.6% HPLC purity. ^1^H-NMR (400 MHz, *DMSO-d_6_*) δ 7.95 (d, *J* = 8.4 Hz, 2H), 7.75 (d, *J* = 8.0 Hz, 1H), 7.58 (t, *J* = 8.0 Hz, 1H), 4.39 (t, *J* = 6.0 Hz, 2H), 3.69 (t, *J* = 6.4 Hz, 2H), 2.31–2.25 (m, 2H) ppm. ^13^C-NMR (101 MHz, *DMSO-d_6_*) δ 165.04, 133.93, 133.59, 132.35, 132.05, 131.32, 129.07, 128.26, 63.04, 31.87, 21.55 ppm. ESI-MS *m/z*: 252.1 [M + H]^+^.

*3-(4-Bromophenyl)-6,7-dihydro-5H-[1,2,4]triazolo[3,4-b][1,3]thiazine* (**5g**) 47% yield, 97.1% HPLC purity. ^1^H-NMR (400 MHz, CDCl_3_) δ 8.15 (t, *J* = 2.0 Hz, 1H), 7.96 (dt, *J* = 8.0 Hz, *J* = 1.2 Hz, 1H), 7.69 (dd, *J* = 8.0 Hz, *J* = 1.2 Hz, 1H), 7.33 (t, *J* = 8.0 Hz, 1H), 4.47 (t, *J* = 6.0 Hz, 2H), 3.54 (t, *J* = 6.4 Hz, 2H), 2.36–2.29 (m, 2H) ppm. ^13^C-NMR (101 MHz, CDCl_3_) δ 165.09, 140.70, 136.07, 132.57, 131.91, 130.04, 128.21, 122.52, 63.15, 31.71, 29.33 ppm. ESI-MS *m/z*: 296.0, 298.0 [M + H]^+^.

*3-(3-Iodophenyl)-6,7-dihydro-5H-[1,2,4]triazolo[3,4-b][1,3]thiazine* (**5h**) 45% yield, 96.5% HPLC purity. ^1^H-NMR (400 MHz, CDCl_3_) δ 8.38 (t, *J* = 1.6 Hz, 1H), 8.02 (dt, *J* = 8.0 Hz, *J* = 1.6 Hz, 1H), 7.92 (dt, *J* = 8.0 Hz, *J* = 1.6 Hz, 1H), 4.49 (t, *J* = 6.0 Hz, 2H), 3.56 (t, *J* = 6.4 Hz, 2H), 2.38–2.32 (m, 2H) ppm. ^13^C-NMR (101 MHz, CDCl_3_) δ 164.93, 141.94, 138.44, 131.89, 130.13, 128.78, 63.14, 31.73, 29.31 ppm. ESI-MS *m/z*: 344.0 [M + H]^+^.

*3-(3-(Trifluoromethyl)phenyl)-6,7-dihydro-5H-[1,2,4]triazolo[3,4-b][1,3]thiazine* (**5i**) 45% yield, 98.5% HPLC purity. ^1^H-NMR (400 MHz, CDCl_3_) δ 8.29 (s, 1H), 8.23 (d, *J* = 7.6 Hz, 1H), 7.83(d, *J* = 7.6 Hz, 1H), 7.60 (t, *J* = 7.6 Hz, 1H), 4.52 (t, *J* = 6.0 Hz, 2H), 3.55 (t, *J* = 6.4 Hz, 2H), 2.39–2.32 (m, 2H) ppm. ^13^C-NMR (101 MHz, CDCl_3_) δ 165.12, 132.83, 131.30, 130.97, 130.85, 129.64, 129.15, 126.52, 124.99, 122.28, 63.32, 31.67, 29.25 ppm. ESI-MS *m/z*: 286.1 [M + H]^+^.

*3-(4-Fluorophenyl)-6,7-dihydro-5H-[1,2,4]triazolo[3,4-b][1,3]thiazine* (**5j**) 53% yield, 97.9% HPLC purity. ^1^H-NMR (400 MHz, CDCl_3_) δ 7.97 (dt, *J* = 8.4 Hz, *J* = 2.0 Hz, 2H), 7.46 (dt, *J* = 8.4 Hz, *J* = 2.0 Hz, 2H), 4.46 (t, *J* = 6.0 Hz, 2H), 3.55 (t, *J* = 6.4 Hz, 2H), 2.35–2.28 (m, 2H) ppm. ^13^C-NMR (101 MHz, CDCl_3_) δ 166.44, 156.81, 156.35, 129.48, 127.22, 125.42, 62.45, 31.90, 29.53 ppm. ESI-MS *m/z*: 236.1 [M + H]^+^.

*3-(4-Chlorophenyl)-6,7-dihydro-5H-[1,2,4]triazolo[3,4-b][1,3]thiazine* (**5k**) 48% yield, 97.4% HPLC purity. ^1^H-NMR (400 MHz, CDCl_3_) δ 7.97 (dt, *J* = 8.4 Hz, *J* = 2.0 Hz, 2H), 7.42 (dt, *J* = 8.4 Hz, *J* = 2.0 Hz, 2H), 4.47 (t, *J* = 6.0 Hz, 2H), 3.54 (t, *J* = 6.4 Hz, 2H), 2.36–2.29 (m, 2H) ppm. ^13^C-NMR (101 MHz, CDCl_3_) δ 165.54, 139.58, 130.99, 128.79, 128.45, 62.95, 31.76, 29.30 ppm. ESI-MS *m/z*: 252.0 [M + H]^+^.

*3-(4-Bromophenyl)-6,7-dihydro-5H-[1,2,4]triazolo[3,4-b][1,3]thiazine* (**5l**) 47% yield, 97.0% HPLC purity. ^1^H-NMR (400 MHz, CDCl_3_) δ 7.88 (d, *J* = 8.4 Hz, 2H), 7.58 (d, *J* = 8.4 Hz, 2H), 4.45 (t, *J* = 6.0 Hz, 2H), 3.53 (t, *J* = 6.4 Hz, 2H), 2.34–2.28 (m, 2H) ppm. ^13^C-NMR (101 MHz, CDCl_3_) δ 164.70, 148.67, 130.81, 130.14, 127.89, 127.27, 61.98, 30.73, 28.34 ppm. ESI-MS *m/z*: 296.0, 298.0 [M + H]^+^.

*3-(4-Iodophenyl)-6,7-dihydro-5H-[1,2,4]triazolo[3,4-b][1,3]thiazine* (**5m**) 40% yield, 97.7% HPLC purity. ^1^H-NMR (400 MHz, *DMSO-d_6_*) δ 7.92 (d, *J* = 8.4 Hz, 2H), 7.74 (d, *J* = 8.4 Hz, 2H), 4.37 (t, *J* = 6.0 Hz, 2H), 3.67 (t, *J* = 6.4 Hz, 2H), 2.29–2.23 (m, 2H) ppm. ^13^C-NMR (101 MHz, *DMSO-d_6_*) δ 165.90, 138.21, 131.27, 129.79, 129.21, 102.28, 62.76, 31.94, 21.24 ppm. ESI-MS *m/z*: 344.0 [M + H]^+^.

*3-(4-(Trifluoromethyl)phenyl)-6,7-dihydro-5H-[1,2,4]triazolo[3,4-b][1,3]thiazine* (**5n**) 40% yield, 98.1% HPLC purity. ^1^H-NMR (400 MHz, *DMSO-d_6_*) δ 8.20 (d, *J* = 8.0 Hz, 2H), 7.92 (d, *J* = 8.0 Hz, 2H), 4.43 (t, *J* = 6.0 Hz, 2H), 3.70 (t, *J* = 6.4 Hz, 2H), 2.32–2.26 (m, 2H) ppm. ^13^C-NMR (101 MHz, *DMSO-d_6_*) δ 165.18, 134.06, 133.17, 130.63, 130.47, 126.29, 122.85, 63.16, 31.88, 31.56 ppm. ESI-MS *m/z*: 286.1 [M + H]^+^.

*3-(4-(Tert-butyl)phenyl)-6,7-dihydro-5H-[1,2,4]triazolo[3,4-b][1,3]thiazine* (**5o**) 43% yield, 98.4% HPLC purity. ^1^H-NMR (400 MHz, CDCl_3_) δ 7.97 (dt, *J* = 8.4 Hz, *J* = 2.0 Hz, 2H), 7.46 (dt, *J* = 8.4 Hz, *J* = 2.0 Hz, 2H), 4.46 (t, *J* = 6.0 Hz, 2H), 3.55 (t, *J* = 6.4 Hz, 2H), 2.35–2.28 (m, 2H), 1.34 (s, 9H) ppm. ESI-MS *m/z*: 274.1 [M + H]^+^.

*3-(2-Methyl-3-nitrophenyl)-6,7-dihydro-5H-[1,2,4]triazolo[3,4-b][1,3]thiazine* (**5p**) 32% yield, 96.6% HPLC purity. ^1^H-NMR (400 MHz, CDCl_3_) δ 7.99 (d, *J* = 7.6 Hz, 1H), 7.87 (d, *J* = 8.0 Hz, 1H), 7.41 (t, *J* = 8.0 Hz, 1H), 4.50 (t, *J* = 6.0 Hz, 2H), 3.54 (t, *J* = 6.0 Hz, 2H), 2.64 (s, 3H), 2.36–2.29 (m, 2H) ppm. ESI-MS *m/z*: 277.1 [M + H]^+^.

*3-(Pyridin-2-yl)-6,7-dihydro-5H-[1,2,4]triazolo[3,4-b][1,3]thiazine* (**5q**) 36% yield, 96.7% HPLC purity. ^1^H-NMR (400 MHz, CDCl_3_) δ 7.83 (dt, *J* = 7.6 Hz, *J* = 1.2 Hz, 1H), 7.71 (dq, *J* = 9.2 Hz, *J* = 1.2 Hz, 1H), 7.46–7.40 (m, 1H), 7.30–7.25 (m, 1H), 4.48 (t, *J* = 6.0 Hz, 2H), 3.55 (t, *J* = 6.0 Hz, 2H), 2.36–2.30 (m, 2H) ppm. ^13^C-NMR (101 MHz, CDCl_3_) δ 165.25, 161.33, 132.20, 130.09, 125.35, 120.19, 116.49, 63.07, 31.73, 29.27 ppm. ESI-MS *m/z*: 219.1 [M + H]^+^.

*3-Benzyl-6,7-dihydro-5H-[1,2,4]triazolo[3,4-b][1,3]thiazine* (**5r**) 38% yield, 96.9% HPLC purity. ^1^H-NMR (400 MHz, *DMSO-d_6_*) δ 7.34–7.24 (m, 5H), 4.14 (t, *J* = 6.0 Hz, 2H), 3.69 (s, 2H), 3.52 (t, *J* = 6.4 Hz, 2H), 2.13–2.07 (m, 2H) ppm. ^13^C-NMR (101 MHz, *DMSO-d_6_*) δ 171.69, 135.47, 134.92, 129.78, 128.80, 127.25, 62.11, 57.59, 40.79, 31.94 ppm. ESI-MS *m/z*: 232.1 [M + H]^+^.

*3-(4-Bromophenyl)-5,6-dihydrothiazolo[2,3-c][1,2,4]triazole* (**5s**) 36% yield, 96.0% HPLC purity. ^1^H-NMR (400 MHz, CDCl_3_) δ 7.86 (d, *J* = 8.8 Hz, 2H), 7.58 (d, *J* = 8.4 Hz, 2H), 4.55 (t, *J* = 6.0 Hz, 2H), 3.57 (t, *J* = 6.0 Hz, 2H) ppm. ^13^C-NMR (101 MHz, *DMSO-d_6_*) δ 165.16, 134.07, 132.51, 131.66, 129.32, 128.17, 65.11, 31.32 ppm. ESI-MS *m/z*: 282.0, 284.0 [M + H]^+^.

*N-(4-((4-(6,7-dihydro-5H-[1,2,4]triazolo[3,4-b][1,3]thiazin-3-yl)phenyl)amino)phenyl)acetamide* (**5t**) 30% yield, 95.6% HPLC purity. ^1^H-NMR (400 MHz, *DMSO-d_6_*) δ 9.59 (s, 1H), 7.96 (d, *J* = 8.4 Hz, 2H), 7.81 (d, *J* = 8.4 Hz, 2H), 7.32 (d, *J* = 8.4 Hz, 2H), 6.57 (d, *J* = 8.4 Hz, 2H), 5.56 (t, *J* = 5.6 Hz, 1H), 4.42 (t, *J* = 6.0 Hz, 2H), 3.20 (q, *J* = 6.0 Hz, 2H), 2.07–2.02 (m, 5H) ppm. ESI-MS *m/z*: 366.1 [M + H]^+^.

*2-(6,7-Dihydro-5H-[1,2,4]triazolo[3,4-b][1,3]thiazin-3-yl)phenol* (**6**) A solution of the 3-(2-methoxyphenyl)-6,7-dihydro-5*H*-[1,2,4]triazolo[3,4-b][1,3]thiazine (**5d**, 200 mg, 0.8 mmol) and boron tribromide (608 mg, 2.4 mmol) in DCM (30 mL) was stirred at ambient temperature for 4 h. The reaction mixture was poured into 10 mL of water and stirred for another 1 h. Then, the organic solvent was removed and extracted with ethyl acetate (3×), and the combined organic layers were concentrated. Next, the residue was purified by column chromatography (PE:EA = 6:1) to give the title compound **6**. 24% yield, 95.6% HPLC purity. ^1^H-NMR (400 MHz, *DMSO-d_6_*) δ 10.50 (s, 1H), 7.83 (dd, *J* = 8.0 Hz, *J* = 1.6 Hz, 1H), 7.53 (td, *J* = 8.0 Hz, *J* = 1.6 Hz, 1H), 7.00–6.93 (m, 2H), 4.41 (t, *J* = 6.0 Hz, 2H), 3.68 (t, *J* = 6.4 Hz, 2H), 2.31–2.25 (m, 2H) ppm. ESI-MS *m/z*: 234.1 [M + H]^+^.

The target compounds **10a** and **10b** were obtained by general procedure 2–4 in turn. Their total yields and characterization data are as follows.

*3-(Furan-2-yl)-6,7-dihydro-5H-[1,2,4]triazolo[3,4-b][1,3]thiazine* (**10a**) 43% yield, 97.6% HPLC purity. ^1^H-NMR (400 MHz, CDCl_3_) δ 7.59 (s, 1H), 7.19 (d, *J* = 3.6 Hz, 1H), 6.53–6.52 (m, 1H), 4.45 (t, *J* = 6.0 Hz, 2H), 3.53 (t, *J* = 6.4 Hz, 2H), 2.35–2.28 (m, 2H) ppm. ESI-MS *m/z*: 208.1 [M + H]^+^.

*3-(Thiophen-2-yl)-6,7-dihydro-5H-[1,2,4]triazolo[3,4-b][1,3]thiazine* (**10b**) 42% yield, 97.0% HPLC purity. ^1^H-NMR (400 MHz, CDCl_3_) δ 7.82 (dd, *J* = 4.0 Hz, *J* = 1.2 Hz, 1H), 7.58 (dd, *J* = 4.8 Hz, *J* = 1.2 Hz, 1H), 7.12 (t, *J* = 4.4 Hz, 1H), 4.45 (t, *J* = 6.0 Hz, 2H), 3.55 (t, *J* = 6.4 Hz, 2H), 2.35–2.28 (m, 2H) ppm. ESI-MS *m/z*: 224.0 [M + H]^+^.

*2-(6,7-Dihydro-5H-[1,2,4]triazolo[3,4-b][1,3]thiazin-3-yl)benzoic acid* (**14**) A mixture of isobenzofuran-1,3-dione (**11**, 593 mg, 0.4 mmol) and hydrazinecarbothioamide (364 mg, 0.4 mmol) in 15 mL of MeCN was warmed to 80 °C and stirred for 4 h. Then, the organic solvent was removed and extracted three times with ethyl acetate, dried and concentrated to give 2-(2-carbamothioylhydrazine-1-carbonyl)benzoic acid (**12**) in 84% yield. Using intermediate **12** as raw material, the target compound **14** was obtained by general procedure 3 and 4 in turn, and the total yield was 35%, 95.8% HPLC purity. ^1^H-NMR (400 MHz, *DMSO-d_6_*) δ 7.81 (d, *J* = 6.0 Hz, 1H), 7.59 (d, *J* = 6.0 Hz, 2H), 7.54 (d, *J* = 6.4 Hz, 1H), 4.27 (t, *J* = 4.8 Hz, 2H), 3.43 (t, *J* = 6.4 Hz, 2H), 2.36 (s, 1H), 2.10 (s, 1H) ppm. ESI-MS *m/z*: 260.1 [M–H]^−^.

*2-(5-Oxo-5,6-dihydrothiazolo[2,3-c][1,2,4]triazol-3-yl)benzoic acid* (**15**) A mixture of 2-(5-thioxo-4,5-dihydro-1H-1,2,4-triazol-3-yl)benzoic acid (**13**, 340 mg, 1.5 mmol) and NaAc (1.01 g, 12.3 mmol) was stirred at room temperature for 0.5 h. Then, 2-bromoacetic acid (1.70 g, 12.3 mmol) was added, and the mixture was heated to 80 °C for 6 h. Subsequently, the organic solvent was removed and the residue was purified by column chromatography (PE:EA = 6:1) to give the desired target compound **15**. The total yield was 27%, and HPLC purity was 95.6%. ^1^H-NMR (400 MHz, *DMSO-d_6_*) δ 14.28 (s, 1H), 7.79–7.66 (m, 4H), 4.77 (m, 2H) ppm. ESI-MS *m/z*: 260.1 [M − H]^−^.

### 3.2. Inhibition Assays

All the target compounds were freshly prepared in 100 mM DMSO stock solutions. All the MBL and SBL enzymes used in this study were obtained from our collaborator, Li’s Laboratory in Sichuan University. Each compound was initially evaluated for inhibitory activity to MBL and SBL enzymes at 100 and 10 μM, by preincubation with the appropriate amount of enzymes in the assay buffer for 10 min, followed by adding the substrate FC5 to initiate the reactions, and monitoring the fluorescence at λex of 380 nm and λem of 460 nm. The IC_50_ values for the compounds (with 10 different concentrations in threefold dilution) were further determined. All determinations were tested in triplicate [12,20,38]. The IC_50_ values were obtained from the plot of activity versus inhibitor concentration by using the GraphPad Prism software.

### 3.3. Molecular Docking Assays

The GOLD and AutoDock Vina programs were used here for molecular docking studies. Compound **5l** was prepared using the AutoDockTools to generate a pdbqt file. The crystal structure of VIM-2 complexed with the inhibitor (2*R*)-2-(4-hydroxyphenyl)-2-[[(2*S*)-2-methyl-3-sulfanyl-propanoyl]amino]ethanoic acid (PDB ID: 5Y6E) was used as the docking template. All the water molecules and solvent molecules were removed. Gasteiger–Marsili charges were added to the protein model, and non-polar hydrogens were then merged onto their respective heavy atoms. The grid center was set as coordinates of the center of the co-crystal inhibitor, and the grid size was 20 Å × 20 Å × 20 Å, which encompasses the whole VIM-2 active site. The docking simulation using GOLD was prepared according to the user guidance. The final binding pose was chosen if it was top ranked by both the docking programs. The binding pose figure was prepared by using the PyMol program.

## 4. Conclusions

In summary, a series of [1,2,4]triazole derivatives were synthesized according to the hit compound **5a**. The preliminary SAR analyses on these synthesized derivatives led to the identification of a number of VIM-2 inhibitors, e.g., **5l** (with an IC_50_ value of 38.36 μM). The selectivity analyses revealed that **5l** may have a selectivity to VIM-2 MBL over other MBLs and SBLs. Compound **5l** was observed by molecular docking to bind with the VIM-2 via the triazole involving zinc coordination and make hydrophobic interactions with residues on the flexible L1 loop. We think that this work will provide a new scaffold to develop MBL inhibitors to combat MBL-mediated β-lactam resistance.

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
