# Peer review of "Discovery of [1,2,4]Triazole Derivatives as New Metallo-β-Lactamase Inhibitors"

_molecules, 2019, doi:10.3390/molecules25010056_

Round 1

Reviewer 1 Report

The manuscript is well written.  This is a short paper that present new potential metallo-beta-lactamase inihibitors. The synthesis of these inhibitors and the measure of their inhibition propensity is presented.

Here are some comments about this manuscript:

The criteria that has been used to select the most promising compounds from Table 1 is not clear for me. How did the authors choose the compounds for which the IC50 value is measured (Fig 4) ? Page 6, line 150: is there an error associated to the IC50 value ? Docking analysis: Could the authors explain the objectives of the docking simulation ? How are detected the interactions between the enzyme and the inhibitor: which tool has been used ? Detecting that manually is not a good approach because having an interaction depends on a specific geometry (distance, angle). On the other hand, there is a positive charge and a partial positive charge close to a pi system (Fig 5): did the authors check if the geometry allows a cation-pi interaction (they argue that there is an interaction with Arg228, but did they check that) ? Are there different poses obtained by the docking simulation ? What is the value of the computed binding energy of the different poses ? Figure 1: the IC50 values are given for each compound, except for IV (Ki value). Please give the IC50 for IV to be coherent.

Author Response

Responses to the Editor and Reviewers:

First, we would like to thank the editor and the reviewers for their constructive comments. We have carefully addressed the reviewers’ concerns and made corrections. Detailed descriptions related to the revisions are given as follows:

Reviewer 1’s Comments, Point 1:

 “The criteria that has been used to select the most promising compounds from Table 1 is not clear for me. How did the authors choose the compounds for which the IC50 value is measured (Fig 4) ?

Answer: Thank the reviewer for this comment. The most promising compounds were selected from Table 1 based on the inhibition rate against the corresponding targets (the inhibition rate >50%). To make this clear, we have added some descriptions in the revised manuscript. Please see Page 8, Line 146-147 in the revised manuscript.

Reviewer 1’s Comments, Point 2:

Page 6, line 150: is there an error associated to the IC50 value ?

Answer: Thank the reviewer for this comment. After careful examination, we think there has no error associated to the IC50 value.

Reviewer 1’s Comments, Point 3:

Docking analysis: Could the authors explain the objectives of the docking simulation ?

Answer: We thank the reviewer for this comment. As described in the manuscript, the possible binding mode of 5l with VIM-2 were obtained by molecular docking analyses. Please see Page 9, Lines 169-178 in the revised manuscript.

Reviewer 1’s Comments, Point 4:

How are detected the interactions between the enzyme and the inhibitor: which tool has been used ?

Answer: Thanks. As described in the legend of Figure 5, the 2D interactions between VIM-2 and 5l were analyzed by using the Discovery Studio Visualizer, and the 3D view of the interactions were shown by using Pymol.

Reviewer 1’s Comments, Point 5:

Detecting that manually is not a good approach because having an interaction depends on a specific geometry (distance, angle). On the other hand, there is a positive charge and a partial positive charge close to a pi system (Fig 5): did the authors check if the geometry allows a cation-pi interaction (they argue that there is an interaction with Arg228, but did they check that) ?

Answer: Thanks. The interactions were analyzed by using the Discovery Studio Visualizer, in which a specific geometry should be considered. There are no cation-pi interactions between 5l and Arg228 detected.

Reviewer 1’s Comments, Point 6:

Are there different poses obtained by the docking simulation ?

Answer: Thanks. There is no obvious differences between the results of GOLD and autodock vina. The pose shown in Figure 5 was ranked first by the docking scores in both programs.

Reviewer 1’s Comments, Point 7:

What is the value of the computed binding energy of the different poses ?

Answer: Thanks. The docking scores for the pose shown in Figure 5 are 53.18 and -7.5 kcal/mol by GOLD and vina programs, respectively. In GOLD program, the goldscores for the top 10 binding poses are from 53.18 to 50.25. In vina program, the vinascores from the top 10 binding poses are from -7.5 kcal/mol to -6.5 kcal/mol

Reviewer 1’s Comments, Point 8:

Figure 1: the IC50 values are given for each compound, except for IV (Ki value). Please give the IC50 for IV to be coherent.

Answer: Thank the reviewer for this comment. The reported activity of compound IV is the Ki value rather than IC50 value, so we only gave the Ki value for IV.

Reviewer 2 Report

Authors describe the preparation of a small series of a series of [1,2,4]triazole derivatives displaying inhibition to the clinically relevant subclass MBLs B1 VIM-2 and MBL. For the best inhibitors micromolar inhibition was detected.

Disapprovingly, Authors applied molecular modeling at the end of their project and not from the beginning, despite the possibility to apply SBBD to reduce the total number of synthesized molecules.

Beside the large amount of synthesized molecules, the activity of the best inhibitor did to really increased from the starting hit. At least not as much as claimed from the Authors.

Going into details:

Pag 1 line 42.

Authors miss to cite important recent works on MBLs from many laboratories around the world. A good choice could be to refer at least to a very recent reviews such as ACS Infect Dis. 2019 Jan 11;5(1):9-34. doi:10.1021/acsinfecdis.8b00247.

Figure 1.

Authors should absolutely include compound taniborbactam, the best-known inhibitor active against SBLs and MBLs and now in clinical trial phase 3. See for these J Med Chem. 2019 Sep 26;62(18):8544-8556. doi: 10.1021/acs.jmedchem.9b00911.

Could be also interesting to refer to one of the inhibitors obtained from Spyrakis et al by virtual screening. See ACS Med. Chem. Lett. 2018, 9, 1, 45-50

Pag 2 line 61

By random screening of our in-house compound library....

It is not clear at all whether the compounds were rationally designed or not. Authors should explain what prompted them to synthesize some derivatives rather than others. Were they selected by serendipity, from available compounds/ known chemistry or by virtual screening? How was compound 5a identified? How where derivatives designed? Which residues in the binding site where meant to target?

Pag 5 line 115

The inhibitory activities of these target compounds were first tested against B1 VIM-2 at

concentrations of 100 μM and 10 μM. Compared with the hit compound 5a, compounds 5b-5i, 6 and 117 14, containing an ortho or meta position substituents on the phenyl ring, showed comparable or slightly lower potency to VIM-2 at 100 μM, except for 5i (52 ± 4 %) and 6 (57 ± 5 %)....

Authors should be more realistic and consider these data as very similar, no really improvement have been detected ....same order of magnitude is reported. Please comment accordingly in the text.

Pag 7, figure 3.

It should be interesting to see 5l inhibition curve in parallel with that of starting compound 5a. This would help to follow results and discussion.

Tests vs SBLs

Authors tested some of the obtained inhibitors towards representative of SBLs from class A, C and D. Authors claim a selective effect of inhibition. Ok but what for? A cross class inhibition would be very well accepted considering that clinical strains coproduce BLs belonging to more than one class and belonging to SBLs and MBL groups. So this result does not add value to the obtained library of derivatives.

Pag 8

Through molecular docking analyses....

Binding mode predictions. Authors propose a model to explain, only for best compound 5l, the affinity vs the targeted MBL. The results explain, with some limitations, the profile of 5l but not the affinity of the other inhibitors presented in the manuscript. Considering all synthesized compound, the fact that only for best inhibitor 5l binding mode was predicted is not sufficient., At least some comments on other derivatives, more and less active, would have been welcomed.

Moreover, worst inhibitor could be used to explain what is good and what is not for activity.

Pag 12 line 357

Each compound was initially evaluated for inhibitory activity to MBL and SBL enzymes

 at 100 μM and 10 μM, by the preincubation with the appropriate amount of enzymes in the assay buffer for 10 min, followed by adding the substrate FC5.

Authors should check the preincubation effect and the time dependency of the inhibition. In other words, did authors considered the possibility of a time dependence inhibition for their inhibitors? (Authors need to evaluate time dependency. you could refer to ChemMedChem. 2018 Apr 6;13(7):713-724. doi: 10.1002/cmdc.201700788)

For in vitro validation, Authors should report for each MBL the concentration of reporter substrate used in the tests and the Km of the reporter S against each BL targeted.

I suggest authors to test their best inhibitor at least against lab strains overexpressing Bls of interest and targeted in their studies, in synergy with a beta-lactam antibiotic.

Data against clinical strains would provide more valuable data. However, considering the low potency of described inhibitors, the choice could be justified since lab strains -respect to the clinical ones - can easier confirm the ability of a molecule to reach the desired site of action (for example the periplasmatic space).

I hope that my comments will help Authors in improving the quality of their presentation/manuscript.

Author Response

Responses to the Editor and Reviewers:

First, we would like to thank the editor and the reviewers for their constructive comments. We have carefully addressed the reviewers’ concerns and made corrections. Detailed descriptions related to the revisions are given as follows:

Reviewer 2’s Comments, Point 1:

Pag 1 line 42. Authors miss to cite important recent works on MBLs from many laboratories around the world. A good choice could be to refer at least to a very recent reviews such as ACS Infect Dis. 2019 Jan 11;5(1):9-34. doi:10.1021/acsinfecdis.8b00247.”

Answer: Thank the reviewer for this comment. According to the reviewer's opinion, we have included the mentioned reference (ACS Infect Dis. 2019 Jan 11; 5 (1):9-34.). Please see Ref 21 in the revised manuscript.

Reviewer 2’s Comments, Point 2:

Figure 1. Authors should absolutely include compound taniborbactam, the best-known inhibitor active against SBLs and MBLs and now in clinical trial phase 3. See for these J Med Chem. 2019 Sep 26; 62(18):8544-8556. doi: 10.1021/acs.jmedchem.9b00911. Could be also interesting to refer to one of the inhibitors obtained from Spyrakis et al by virtual screening. See ACS Med. Chem. Lett. 2018, 9, 1, 45-50”

Answer: Thank the reviewer for this comment. According to the reviewer's opinion, we have included the mentioned references (J Med Chem. 2019 Sep 26; 62(18):8544-8556; ACS Med. Chem. Lett. 2018, 9, 1, 45-50). Please see Ref 22 and 23 in the revised manuscript.

Reviewer 2’s Comments, Point 3:

Pag 2 line 61: By random screening of our in-house compound library....It is not clear at all whether the compounds were rationally designed or not. Authors should explain what prompted them to synthesize some derivatives rather than others. Were they selected by serendipity, from available compounds/ known chemistry or by virtual screening? How was compound 5a identified? How where derivatives designed? Which residues in the binding site where meant to target?”

Answer: Thank the reviewer for this comment. Actually, we have an in-house library containing more than 1000 compounds, which were synthesized for other targets such as IDO/TDO and sirtuins or for synthetic methodology studies. Herein, we just screened this databases, and identified that 5a showed activity against VIM-2, and its chemical scaffold has not been reported as MBL inhibitors. So we further synthesized new derivatives and investigated the structure-activity relationships.

Reviewer 2’s Comments, Point 4:

Pag 5 line 115: The inhibitory activities of these target compounds were first tested against B1 VIM-2 at concentrations of 100 μM and 10 μM. Compared with the hit compound 5a, compounds 5b-5i, 6 and 14, containing an ortho or meta position substituents on the phenyl ring, showed comparable or slightly lower potency to VIM-2 at 100 μM, except for 5i (52 ± 4 %) and 6 (57 ± 5 %)....Authors should be more realistic and consider these data as very similar, no really improvement have been detected ....same order of magnitude is reported. Please comment accordingly in the text.”

Answer: Thanks. Although we synthesized 25 compounds, no significant improvement for activity was achieved. Probably, this scaffold may not be a good metal-binding motif and may not be suitable to develop highly potent compounds for MBLs.

Reviewer 2’s Comments, Point 5:

Pag 7, figure 3. It should be interesting to see 5l inhibition curve in parallel with that of starting compound 5a. This would help to follow results and discussion.”

Answer: Thanks. As described in the manuscript, the structure-activity relationship data identified 5l as the most potent compound, which has a slightly better potency than the starting compound 5a. The molecular docking analyses may indicate the possible binding mode for 5l, which has been included in the manuscript. 

Reviewer 2’s Comments, Point 6:

Tests vs SBLs Authors tested some of the obtained inhibitors towards representative of SBLs from class A, C and D. Authors claim a selective effect of inhibition. Ok but what for? A cross class inhibition would be very well accepted considering that clinical strains coproduce BLs belonging to more than one class and belonging to SBLs and MBL groups. So this result does not add value to the obtained library of derivatives.”

Answer: Thanks. Actually, we here used SBLs for counter screening to indicate the specific inhibition to MBLs.

Reviewer 2’s Comments, Point 7:

Pag 8Through molecular docking analyses....Binding mode predictions. Authors propose a model to explain, only for best compound 5l, the affinity vs the targeted MBL. The results explain, with some limitations, the profile of 5l but not the affinity of the other inhibitors presented in the manuscript. Considering all synthesized compound, the fact that only for best inhibitor 5l binding mode was predicted is not sufficient., At least some comments on other derivatives, more and less active, would have been welcomed. Moreover, worst inhibitor could be used to explain what is good and what is not for activity.”

Answer: Thanks. This is a good suggestion. According to the docking poses, we can explain why 5i, 5o, 5t, and 6 showed lower potency to VIM-2 (probably because they fit not so good with Tyr67 and Arg228).

Reviewer 2’s Comments, Point 8:

Pag 12 line 357: Each compound was initially evaluated for inhibitory activity to MBL and SBL enzymes at 100 μM and 10 μM, by the preincubation with the appropriate amount of enzymes in the assay buffer for 10 min, followed by adding the substrate FC5. Authors should check the preincubation effect and the time dependency of the inhibition. In other words, did authors considered the possibility of a time dependence inhibition for their inhibitors? (Authors need to evaluate time dependency. you could refer to ChemMedChem. 2018 Apr 6; 13(7):713-724. doi: 10.1002/cmdc.201700788)”

Answer: Thanks. This is a good comment. Indeed, we used pre-experiments to examine the time-dependent inhibition. These inhibitors have no time-dependent inhibition to VIM-2 because they are common reverse substrate-competitive inhibitors as indicated by the molecular docking analyses.

Reviewer 2’s Comments, Point 5:

For in vitro validation, Authors should report for each MBL the concentration of reporter substrate used in the tests and the Km of the reporter S against each BL targeted.”

Answer: Thanks. All the MBL and SBL enzymes used in this study were obtained from our collaborator, Li’s Laboratory in Sichuan University. All the assay conditions were used as same as they used. The Km values of FC-5 with all BLs were well-studied and reported (please see Wang Y-L, et al, J. Med. Chem. 2019; 62: 7160-7184; Liu et al, Eur. J. Med. Chem. 2018; 145: 649-660)

Round 2

Reviewer 2 Report

Dear Authors.

I do not feel like that the suggestions have been really considered.

Figure 1 still does not show taniborbactam.

Also for point 5, 6 and 7 and 9. My suggestions have been commented in the answer to referee but no changes in the text have been introduced. expecially for 5 and 7.

Author Response

Responses to the Editor and Reviewers:

First, we would like to thank the editor and the reviewers for their constructive comments. We have carefully addressed the reviewers’ concerns and made corrections. Detailed descriptions related to the revisions are given as follows:

Reviewer 2’s Comments, Point 1:

 “Inclusion of taniborbactam in Figure 1 (point 1) and 5a inhibition curve in Figure 3 (point 5) can be done with little extra effort

Answer: Thank the reviewer for this comment. According to the reviewer's suggestion, we have included tanniborbactam in Figure 1, and corresponding descriptions in the introduction section. Details please see Page 2, line 50-53, and Figure 1 in the revised manuscript.

The inhibition curve of 5a to VIM-2 and the IC50 value were included. Please see Page 2, Line 66 in the revised manuscript and Figure S1 in the revised supporting information.

Reviewer 2’s Comments, Point 2:

A more detailed discussion based on docking results could also be provided in the text, with supporting material provided as Supplementary Information if needed. (point 7)

Answer: Thanks. We have included more details regarding the docking results in the revised manuscript. Please see Page 9, Lines 174-178 in the revised manuscript.

Reviewer 2’s Comments, Point 3:

Appropriate comments relative to points 6 and 9 could also be added to the main text

Answer: Thanks. We included the appropriate descriptions relative to points 6 and 9 in the revised manuscript. Please see Page 7 Lines 140-141 and Page 8 Lines 167-168 in the revised manuscript.